# Protective Effect of *Amaranthus cruentus* L. Seed Oil on UVA-Radiation-Induced Apoptosis in Human Skin Fibroblasts

**DOI:** 10.3390/ijms241310795

**Published:** 2023-06-28

**Authors:** Katarzyna Wolosik, Magda Chalecka, Jerzy Palka, Arkadiusz Surazynski

**Affiliations:** 1Independent Cosmetology Laboratory, Medical University of Bialystok, Kilinskiego 1, 15-089 Bialystok, Poland; katarzyna.wolosik@umb.edu.pl; 2Department of Medicinal Chemistry, Medical University of Bialystok, Kilinskiego 1, 15-089 Bialystok, Poland; magda.chalecka@umb.edu.pl (M.C.); pal@umb.edu.pl (J.P.)

**Keywords:** UVA radiation, oxidative stress, dermal fibroblasts, apoptosis, antioxidant, *Amaranthus cruentus* L. seed oil, sun-protective substance, pharmacy, cosmetology

## Abstract

Since the exposure of fibroblasts to prolonged UVA radiation induces oxidative stress and apoptosis, there is a need for effective skin protection compounds with cytoprotective and antioxidant properties. One of their sources is *Amaranthus cruentus* L. seed oil (AmO), which is rich in unsaturated fatty acids, squalene, vitamin E derivatives and phytosterols. The aim of this study was to evaluate whether AmO evokes a protective effect on the apoptosis induced by UVA radiation in human skin fibroblasts. UVA radiation at an applied dose of 10 J/cm^2^ caused a significant reduction in the survival of human skin fibroblasts and directed them into the apoptosis pathway. Increased expression of p53, caspase-3, caspase-9 and PARP proteins in UVA-treated fibroblasts suggests the intrinsic mechanism of apoptosis. Application of the oil at 0.1% and 0.15% concentrations to UVA-treated cells decreased the expression of these proteins, which was accompanied by increased cell survival. Similarly, the UVA-dependent decrease in the expression of p-Akt and mTOR proteins was restored under the effect of the studied oil. The molecular mechanism of this phenomenon was related to the stimulation of antioxidant processes through the activation of Nrf2. This suggests that AmO stimulated the antioxidant system in fibroblasts, preventing the effects of UVA-induced oxidative stress, which may lead to pharmaceutical and cosmetological applications as a sun-protective substance.

## 1. Introduction

Sunlight is electromagnetic radiation with a full range of wavelengths from ultraviolet (UV) radiation (UVA, UVB and UVC in the range of 200–400 nm) to visible radiation (in the range of 400–800 nm) and infrared (IR) radiation (IRA, IRB and IRC in the range of 800–15,000 nm). Among all types of radiation, ultraviolet radiation has the most significant effect on the human body. Seventy percent of UVB radiation that reaches the skin is absorbed by the stratum corneum, 20% goes to the spinosum and basal layer of the epidermis, and only 10% penetrates the papillary layer of the skin dermis [1]. UVA radiation is partially absorbed by the epidermis, but as much as 20–30% reaches the deep epidermal reticular layer of the dermis [2]. Unlike UVB radiation, UVA radiation, due to its high penetration properties, reaches deeper parts of the skin and affects dermal compartments. When exposed to UVA radiation at a dose of 25 J/cm^2^, dermal fibroblasts located in the upper layer of the skin die within 48 h of exposure through an apoptotic process. In contrast, the structure and organization of the epidermis does not change significantly. Studies confirmed that dermal fibroblasts are more sensitive to UVA-induced oxidative stress than keratinocytes [3,4]. UV radiation generates free radicals that are responsible for cellular stress and DNA damage, which leads to the activation of the p53 tumor suppressor protein. Activation of the p53 protein in irradiated cells leads to cell cycle arrest to allow for the repair of DNA damage. However, when the damage is extensive or repair is impossible, it induces apoptotic cell death via the intrinsic (mitochondrial) caspase-9-dependent pathway of apoptosis. Activated caspase-9 then cleaves and activates effector caspases, such as caspase-3 and -7, which carry out apoptosis [5,6,7]. UV also negatively affects the AKT/mTOR signaling pathway, which is closely related to cell cycle arrest and apoptosis. The AKT/mTOR signaling pathway can antagonize p53 activity in response to UV radiation. Shifting the balance between p53 function and AKT/mTOR signaling may determine cell death or survival [6]. Due to exposure to damaging UV radiation, the cells of the different layers of the skin, namely, epidermal keratinocytes and dermal fibroblasts, activate several protection mechanisms. They include the high activity of repair and antioxidant enzymes, a large pool of non-enzymatic antioxidants and redox-sensitive transcription factors, including Nuclear factor-erythroid 2 related actor 2 (Nrf2), which induces the expression of cytoprotective proteins [8]. These processes become depleted during prolonged UV exposure; therefore, the search for effective skin-protective compounds with cytoprotective and antioxidant properties is required. Seed oils containing free fatty acids, vitamins, sterols, carotenoids and phenols have emerged as promising sources of such compounds [5].

Seed oil from *Amaranthus cruentus* L. (AmO) is one of the potential sources of such a compound, as it contains abundant active substances. Among them, linoleic acid is well-known as an antioxidant [9]. The species *Amaranthus cruentus* L. from which the above-mentioned oil is extracted belongs to the *Caryophyllales* order, *Amaranthaceae* family, *Amaranthoideae* subfamily and the *Amaranthus* genus. Amaranthus seeds exhibit a notable chemical composition characterized by a substantial content of proteins, peptides and amino acids. The abundance of proteins and amino acids in amaranth confers medicinal benefits, including cholesterol-lowering, antioxidant, anticancer, anti-allergic and antihypertensive properties. The oil extracted through cold-pressing of the grains yields a relatively low lipid content of only 7–8%. However, these lipids possess high value due to the presence of unsaturated fatty acids, tocopherols, tocotrienols, phytosterols and squalene, which are not concurrently found in comparable common oils. Of particular significance is the presence of squalene. Among the fatty acids present in *Amaranthus* seeds, linoleic acid (C18:2) stands out as the predominant component comprising approximately 47% of the total fatty acid content [9,10]. Oleic acid (C18:1) was detected at a concentration of 24%, while palmitic acid (C16:0) accounted for 23.45% of the total fatty acids. Similar findings of lower palmitic acid and higher oleic acid contents were reported by Gamel et al. [9,11]. The content of stearic acid (C18:0) was determined to be 4.16% of the total fatty acids. Additionally, α-linolenic acid exhibited a content of 0.69% of the total fatty acids [9,10]. Berganza et al. [12] conducted an analysis of squalene content in seeds and found it to be between 3.20% and 5.80% of the total fatty acids. Squalene, which is known for its potent biological activity, exhibits exceptional antioxidant properties and acts as a natural emollient, effectively reducing transepidermal water loss and enhancing skin hydration. These properties make it a valuable ingredient in formulations for the treatment of skin conditions, such as atopic dermatitis and psoriasis. Additionally, squalene was found to have a beneficial impact on acne vulgaris, inhibiting the development of acne lesions [13].

However, AmO has an unknown effect on UVA-treated cells. Since the oil is rich in the described compounds, it may also evoke protective activity against oxidative stress induced by prolonged exposure to sunlight. The result of this study may lead to potential pharmaceutical and cosmetological applications of AmO as a source of sun-protective agents. Thus, the objective of this study was to assess the protective effect of AmO on UVA-stimulated apoptosis in a cellular model of human skin fibroblasts.

## 2. Results and Discussion

The skin, being the largest organ of the human body, has the most direct contact with external factors, especially sunlight. Prolonged exposure to UV radiation causes photoaging, which is superimposed on aging caused by the passage of time (chronological aging). During skin exposure, UV energy is absorbed by endogenous cellular chromophores with their excitation, which occurs in the presence of molecular oxygen, producing several oxidation products and reactive oxygen species (ROS). This induces redox imbalance, which impedes collagen synthesis; causes DNA, proteins and lipid damage; disrupts biological membrane structures; and modulates fibroblast activity [14].

Therefore, the effect of UVA radiation on the above processes was studied in human skin fibroblasts. The cells were exposed to UVA at different doses, as shown in Figure 1A. It was found that the fibroblast viability decreased with increasing UVA dose. Doses of 2 J/cm^2^ and 5 J/cm^2^ did not significantly affect the viability of the cells. Doses of 6 J/cm^2^ and 10 J/cm^2^ reduced the fibroblast viability to about 78% and 54% of the control value, respectively. Doses of 12 J/cm^2^, 15 J/cm^2^ and 20 J/cm^2^ reduced the fibroblast viability to about 38%, 29% and 6% of the control value, respectively. Based on the experiment, the dose of 10 J/cm^2^, contributing to about an IC_50_ value of cell viability, was selected for further experiments.

Due to its unique chemical composition, AmO exhibits cytoprotective and antioxidant potential, suggesting its promising application as a UV-radiation-protective substance for skin care purposes [15]. We found that the effect of AmO on cell viability varied depending on the concentration used. The application of low concentrations (0.05%, 0.1% and 0.15%) of the oil had no significant effect on the viability of human skin fibroblasts. In contrast, the use of higher concentrations (0.2% and 0.25%) of this oil resulted in a decrease in cell viability to about 58% and 57% of the control value, respectively. Studies by Semen et al. [16] provided evidence that amaranth oil exhibits both antioxidant and pro-oxidant effects in human lung fibroblasts. At low concentrations of the oil, fibroblasts were protected from oxidative stress, while incubation with high concentrations of the oil resulted in an increase in intracellular ROS generation and cell damage. The observed decrease in cell viability at high concentrations of AmO was attributed to the physical effect of the oil itself on the cells, which led to their exclusion from further experiments (Figure 1B).

Our studies indicated a cytotoxic effect of UVA radiation on human skin fibroblasts and its involvement in the generation of significant amounts of ROS. Considering that ROS are suppressors of basic cellular functions, as well as pro-apoptotic factors [17], we investigated the protective–antioxidant potential of AmO at selected concentrations in UVA-treated human skin fibroblasts. We found that UVA radiation generated significant amounts of ROS in the studied cells. Treatment with a 0.05% concentration of AmO reduced the formation of ROS but not as significantly as at concentrations of 0.1% and 0.15% (Figure 2). Gegotek et al. [8] observed a decrease in ROS levels in fibroblasts exposed to UVA radiation, which was also evident in the presence of other oils, including sea buckthorn seed oil. Treatment with sea buckthorn seed oil prevented UVA-induced redox imbalance in fibroblasts. It was found that the incubation of fibroblasts with sea buckthorn oil caused a decrease in ROS generation by about 25% [8,18].

Based on the results obtained in the MTT test, we observed that the survival rate of cells treated with UVA at a dose of 10 J/cm^2^ was decreased by 45% and this was a statistically significant difference compared with the control (*p* < 0.05). Comparing the effect of UVA on cells in the presence of AmO at 0.05%, 0.10% and 0.15% concentrations, we observed that the cell survival was 13%, 19% and 23% higher, respectively, than in the UVA-treated cells and these effects were statistically significant (*p* < 0.05). These data indicate the antioxidant effect of AmO at selected concentrations in UVA-treated fibroblasts (Figure 3, Table 1).

The MTT experiment results suggest that UVA irradiation of human skin fibroblasts impaired their survival, while the presence of AmO reversed the cytotoxic effect of UVA radiation on this process. However, the mechanism of this effect is unknown. Cytometric assays for apoptosis and cell survival were performed using a Nucleo-Counter^®^ NC-3000^TM^ imaging cytometer. The survival rate of fibroblasts treated with AmO, regardless of the concentration used, remained at 82–86%. In cells irradiated with UVA at a dose of 10 J/cm^2^, a negative effect of this radiation on fibroblast survival was noted, which was about 56% of the control. AmO at concentrations of 0.05%, 0.1% and 0.15% counteracted the toxic effect of UVA on the cells, whose survival rates were reduced but to lower levels, i.e., to about 68%, 75% and 79% of the values obtained in the UVA-treated group, respectively (Figure 4).

Redox balance is essential for the physiological function of the cell. Under physiological conditions, the cytosol has a highly reducing environment due to its high level of reduced glutathione and a complex redox enzyme system that maintains glutathione in a reduced state. In most eukaryotic cells, the cysteine thiols of proteins are the most sensitive to changes in the cellular redox state and 98% of thiols remain in a reduced form. Oxidative stress contributes to the non-selective and irreversible oxidation of protein thiols, glutathione depletion and cell death. Thus, the content of reduced thiols is an indicator of ROS formation and apoptosis [16]. A study by Emonet et al. [19] indicated that the UVA sensitivity of human skin fibroblasts is associated with a decrease in their intracellular glutathione (GSH) level. In our studies, we found that AmO alone caused a slight decrease in the level of reduced thiols, which remained at 78–88% of the control. In contrast, in the UVA-irradiated cells, a negative effect of this radiation on the level of reduced thiols was noticed, which was about 62% of the control. Comparing the effect of UVA on the cells cultured in the presence of AmO at concentrations of 0.05%, 0.1% and 0.15%, the level of reduced thiols increased to about 79%, 85% and 87% of the control, respectively (Figure 5). This suggests that the tested oil had a protective effect on the UVA-induced disturbances in the oxidation–reduction balance in the fibroblasts.

Studies by Bernard et al. [20] indicated that in contrast to UVB, UVA radiation at doses consistent with realistic sun exposure induces oxidative stress in cells. UVA produces singlet oxygen, which depolarizes mitochondrial membranes, induces oxidative damage to DNA and initiates late apoptosis. In contrast, UVB causes only late apoptosis. Interestingly, exposure to UVA in vivo did not induce apoptosis in epidermal cells. These results were confirmed in reconstructed UVA-treated skin, in which apoptosis was observed in fibroblasts of the dermis but not in epidermal keratinocytes [21]. Our study indicated that UVA radiation at a dose of 10 J/cm^2^ significantly reduced the survival rate and induced apoptosis in the fibroblasts. The application of AmO at concentrations of 0.1% and 0.15% significantly reduced these processes (Figure 5). We also demonstrated that the used dose of UVA radiation induced the mitochondrial apoptosis pathway, which was accompanied by a significant increase in the expression of the apoptosis-inducing factor p53 protein (Figure 6), as well as caspase-9 (Figure 7) and caspase-3 (Figure 8), which play a key role in this process. Treatment of the cells with selected concentrations of *AmO* caused decreases in the expressions of these proteins, which are the markers of apoptosis. This suggests that UVA-dependent activation of apoptosis through the intrinsic pathway is inhibited by AmO in fibroblasts.

To maintain the integrity of the skin barrier, cells respond to UV by engaging other mechanisms by which they can survive under stress conditions. Among them are AKT/mTOR and p53 signaling as potential regulators of the life and death of irradiated cells [19]. Our study indicated a decrease in the expression of AKT and mTOR proteins under UVA radiation. The expression of the p53 protein was significantly increased. The application of AmO reversed the negative effects of UVA irradiation on this pathway. It showed the strongest effect at 0.1% and 0.15% (Figure 9 and Figure 10). The UVA-reduced expression of p-AKT and mTOR proteins was restored by AmO and accompanied by increased cell survival and reduced p53 expression, suggesting a pro-survival mechanism of the tested oil.

The ROS lead to the formation of oxidation-dependent DNA damage, such as single-strand breaks. UVA-induced DNA damage also leads to the mutations of proto-oncogenes and tumor suppressor genes, such as the previously described p53 [20]. AmO evoked a protective effect on the process of DNA biosynthesis in UVA-treated fibroblasts. In the UVA-treated cells, DNA biosynthesis was decreased by 45% compared with the control cells. In the presence of AmO at 0.05%, 0.10% and 0.15% concentrations, DNA biosynthesis was increased by 12%, 29% and 31%, respectively, despite the toxic effect of UVA and this was statistically significant (*p* < 0.05) (Figure 11, Table 2). Based on the obtained results, it can be assumed that the application of AmO protected cellular DNA from the damaging effects of UVA radiation.

UV-induced DNA strand breaks initiate apoptosis. Poly(ADP-ribose) polymerase (PARP) enzymes constitute a large family of 18 proteins. They represent enzymes that are activated immediately after DNA strand breaks, and thus, the activity may be a marker of DNA damage [20,22]. We observed that UVA radiation at the applied dose increased the expression of PARP. The use of a 0.05% concentration of AmO did not prevent increased PARP expression after UVA irradiation, while the use of this oil at 0.1% and 0.15% concentrations markedly reduced this process (Figure 12). This shows that PARP proteins were involved in the mechanism of apoptosis that occurred as a result of this radiation. Therefore, the reduction in PARP protein expression under the influence of the applied concentrations of AmO indicates its protective effect against the damaging effects of UVA radiation on DNA.

In this report, we provide evidence that UVA-treated fibroblasts impaired DNA biosynthesis and reduced survival through the activation of apoptosis, while the addition of AmO evoked a protective effect on these processes. The molecular mechanism of this phenomenon is related to the activation of transcription factor Nrf2 [23]. Translocation of the Nrf2 into the cell nucleus under oxidative stress conditions activates the expression of numerous cytoprotective genes that enhance cell survival. These include detoxification and antioxidant enzymes [20]. In this study, we found an increased expression of Nrf2 and its translocation to the nucleus in UVA-irradiated human skin fibroblasts; however, treatment with AmO enhanced this expression (Figure 13). The above data suggest that the studied oil stimulated the antioxidant system in the fibroblasts and counteracted the effects of UVA-induced oxidative stress through the activation of the Nrf2-dependent pathways. Similar results were observed for sea buckthorn seed oil. Treatment of UV-irradiated skin cells with this oil promoted antioxidant activity in the cells through Nrf2 activation [8].

## 3. Materials and Methods

### 3.1. Materials

#### Plant Materials

The *Amaranthus cruentus* L. seed oil used in this research was a validated, commercially available product (Szarłat M. and W. Lenkiewicz s.j. Zawady, Poland). It was an unrefined oil obtained via a seed cold-pressing process from a locally cultivated *Amaranthus cruentus* species. The fatty acid profile was as follows: palmitic acid 17.5–19.2%, oleic acid 17.5–20.5%, linoleic acid 52–55%, stearic acid 4.5–5.5%; the oil also contained squalene 6–8% and vitamin E 8–10 mg/100 g. The comprehensive safety data sheet (SDS) for the investigated *Amaranthus cruentus* L. seed oil, which was provided by its manufacturer Szarłat, can be found in the Appendix A.

### 3.2. Methods

#### 3.2.1. Cell Lines and Culture

Human dermal fibroblasts were obtained from the American Type Culture Collection (Manassas, VA, USA); maintained in Dulbecco’s Modified Eagle Medium (DMEM) (PANTM BIOTECH, Aidenbach, Germany); supplemented with 10% Fetal Bovine Serum (FBS) (Gibco, Waltham, MA, USA), 50 U/mL of Penicillin (Pen) (Gibco, Waltham, MA, USA) and 50 μg/mL of Streptomycin (Strep) (Gibco, Waltham, MA, USA); and incubated at 37 °C in 5% CO_2_. The cells were grown on 100 mm dishes in 10 mL of complete medium. The cell culture medium was changed 2–3 times per week. The cells were used after 8–10 passages. For the experiments with the investigated oil, we used DMEM without FBS and Pen/Strep.

#### 3.2.2. Cell Viability Test MTT

To evaluate the cytotoxicity of the UVA radiation and *Amaranthus cruentus* L. seed oil on the fibroblasts, methyl thiazolyl tetrazolium (MTT) salt was used, as described in Carmichael’s method [24]. This method is based on the conversion of yellow tetrazolium bromide MTT solution to the purple formazan derivatives in the live cells, which is due to the activity of the mitochondrial dehydrogenases. For this assay, the cells were cultured in 6-well plates at an initial cell density of 3 × 10^5^ cells/well. When the cells reached about 80% confluency, the culture media were removed. The experiments were undertaken in triplicates. The AmO was used in the following concentrations: 0.05%, 0.1%, 0.15%, 0.2% and 0.25%. After 30 min of incubation, the cell culture media containing the studied oil were removed and the plates were washed with PBS. Subsequently, the cells were exposed to UVA radiation using a Bio-Link Crosslinker BLX 365 (Vilber Lourmat, Eberhardzell, Germany) at various doses, namely, 2 J/cm^2^, 5 J/cm^2^, 6 J/cm^2^, 10 J/cm^2^, 12 J/cm^2^, 15 J/cm^2^ and 20 J/cm^2^ (365 nm) in 1 mL of cold PBS (4 °C). After irradiation, PBS was exchanged for fresh DMEM. The cells were incubated for 24 h. After incubation, the cell culture media were removed and the plates were washed twice with prewarmed PBS. Furthermore, the cells were incubated at 37 °C for 1 h with MTT dissolved in PBS (0.5 mg/mL) in a volume of 1.0 mL per well. When the incubation step ended, the MTT was removed and the formazan derivatives were dissolved in DMSO (1.0 mL per well). To quantify the formed formazan, the absorbance at 570 nm was measured using a spectrophotometer. The viability of the cells treated with the studied oil and UVA irradiation was calculated as a percent of the control value.

#### 3.2.3. ROS Formation

The cells were cultured on black wells in a 96-well plate at 1 × 10^3^ cells/well. When the cells reached 70–80% confluency the culture media were removed, the plate was washed with PBS and 100 µL of medium containing the studied AmO was added into the well. After 30 min, cells were washed with cold PBS (4 °C) and exposed to UVA radiation (Bio-Link Crosslinker BLX 365, Vilber Lourmat, Eberhardzell, Germany) at a total dose of 10 J/cm^2^ (365 nm) in 1 mL of cold PBS (4 °C). Next, the PBS was replaced with DMEM. After 4 h of incubation, 0.5 µM of 2′,7′-dichlorofluorescin diacetate was added to the wells and incubated for 15 min at 37 °C in 5% CO_2_. After incubation, the culture media with DCFDA were removed and the cells were washed twice with prewarmed PBS. Then, the wells were loaded with 100 µL of PBS. The cells were visualized with the BD Pathway 855 Bioimaging system in an environmental control chamber (37 °C in 5% CO_2_) with λ_ex_ = 488 nm and λ_em_ = 521 nm. In its basic state, DCFDA is a nonfluorescent compound, and when oxidized by ROS to DCF, it becomes highly fluorescent.

#### 3.2.4. Cytometric Assay for Apoptosis

The cells were cultured in 6-well plates at an initial density of 3 × 10^5^ cells/well. When the cells reached about 70–80% confluency, the culture media were removed. The wells were washed with PBS, and fresh DMEM containing the studied AmO was added and incubated for 30 min. Next, cells were exposed to UVA irradiation in cold PBS. Then, the PBS was replaced with fresh DMEM in the wells. After 24 h of incubation, the cells (including floating cells) were collected via trypsinization in 1.5 mL Eppendorf tubes. The cells were centrifuged at 800× *g* for 5 min. The supernatant was removed and the cell pellet was resuspended in 100 μL Annexin V binding buffer, and then 2 μL Annexin V-CF488A conjugate was added. In the next step, 2 μL Hoechst 33342 (final concentration: 10 μg/mL) was added and the suspension was gently mixed via pipetting. After that, the cells were incubated at 37 °C for 15 min using a heating block. Next, the stained cells were centrifuged at 400× *g* for 5 min at room temperature and the supernatant was removed. In the last step, the cell pellet was resuspended in 100 μL Annexin V binding buffer supplemented with 10 µg/mL PI and then immediately analyzed in the NC-Slide A2 using a NucleoCounter NC-3000^TM^ (ChemoMetec, Lillerød, Denmark). Cells with low fluorescence intensities of PI (PI negative) and low fluorescence intensities of Annexin V represented living cells with high viability. The cells with high fluorescence intensities of PI (PI positive) and low fluorescence intensities of Annexin V represented early apoptotic cells. The cells with high fluorescence intensities of PI (PI positive) and high fluorescence intensities of Annexin V represented late apoptotic/dead cells.

#### 3.2.5. Free Thiol Groups Assay

The free thiol groups were measured using a NucleoCounter NC-3000^TM^ (ChemoMetec, Lillerød, Denmark). For this assay, the cells were cultured in 6-well plates. When the cells reached 90% confluency, an AmO and UVA irradiation assay was performed (as described in Section 3.2.2). After 24 h of incubation, the cells were collected via trypsinization to 1.5 mL Eppendorf tubes. The cells were centrifuged at 1600× *g* for 10 min. The supernatant was removed and the cell pellet was resuspended in 190 µL of PBS, and 10 µL of Solution 5 (Acridine Orange, Propidium Iodide, and VB-48) was added. The cells were suspended with staining dyes at 10 µL per each chamber of an 8-well NC-Slide A8^TM^ and analyzed using a Vitality Protocol (VitaBright-48^TM^ protocol). Cells with low fluorescence intensities of PI (PI negative) and high fluorescence intensities of VB-48 represented living cells with high viability. The cells with high fluorescence intensities of PI (PI positive) represented dead cells.

#### 3.2.6. DNA Biosynthesis Assay

The DNA biosynthesis was determined using radioactive [methyl-^3^H]thymidine incorporated into the DNA. For this assay, the cells were cultured in 6-well plates. When the cells reached 90% confluency, an AmO and UVA irradiation assay was performed (as described in Section 3.2.2). Straight after the irradiation, 10 µL of 0.5 μCi/mL [methyl-^3^H]thymidine was added to each well and then incubated for 24 h at 37 °C in 5% CO_2_. The experiments were performed in triplicates. The radioactivity of the samples was measured using the Tri-Carb 2810 TR Scintillation Analyzer (PerkinElmer, Waltham, MA, USA). To calculate the DNA biosynthesis, the dpm parameter value (radioactive thymidine incorporated into the DNA) of the treated cells was compared with that of the control group.

#### 3.2.7. Western Immunoblot Analysis

For the analysis of the protein expression via Western blotting, the cells were cultured in 100 mm plates at about 2.0 × 10^6^ cells and when they reached about 70–80% confluency, the culture medium was removed and the cells were UVA-irradiated and treated with AmO (as described in Section 3.2.2). After 24 h of incubation, the culture media were removed and the cells were harvested using a cell lysis buffer supplemented with a protease/phosphatase inhibitor cocktail. The protein concentrations of the samples were determined via the Lowry method [25]. Then, the proteins were separated using the SDS-PAGE method described by Laemmli [26]. After this step, the gels were washed in cold Towbin buffer (25 mM Tris, 192 mM glycine, 20% (*v*/*v*) methanol, 0.025–0.1% SDS, pH 8.3). The proteins in the gels were transferred onto the 0.2 µm nitrocellulose membranes using a Trans-Blot (BioRad, Hercules, CA, USA). The transfer conditions were 200 mA, 3 h in freshly prepared Towbin buffer and the temperature was maintained around 4–8 °C. The blocking of the membranes was performed using 5% NFDM for 1 h at RT. When the blocking was complete, the membranes were washed three times with 20 mL of TBS-T (20 mM Tris, 150 mM NaCl and 0.1% Tween^®^ 20). After the washing step, the membranes were incubated with primary antibodies overnight at 4 °C. The concentration of the primary antibodies was 1:1000. Furthermore, the membranes were washed three times with 20 mL of TBS-T and a secondary antibody conjugated with HRP solutions (1:3000) in 5% NFDM was used for 1 h at RT. Then, the membranes were washed 3 times with 20 mL of TBS-T and visualized using BioSpectrum Imaging System UVP (Ultra-Violet Products Ltd., Cambridge, UK).

#### 3.2.8. Immunofluorescence Staining and Confocal Microscopy

The cells were cultured on a black 96-well plate. When the cells reached 80% confluency, an AmO was added and UVA irradiation was performed (as described in Section 3.2.2). After 24 h, the culture media were removed and the cells were fixed with a 3.7% formaldehyde solution at room temperature for 10 min. Then, the plate was washed once with 100 µL/well of PBS. Later, the permeabilization with a 0.1% Triton X-100 solution and a 10 min step was performed. After the permeabilization the plate was washed twice with PBS and 3% FBS was used as a blocking agent at room temperature for 30 min. After the FBS removal, 50 µL of the primary antibody (1:50), diluted in 3% FBS, was added and the plate was incubated for 1 h at room temperature. After the incubation, the primary antibody plate was washed three times with PBS. Then, 50 µL per well of the secondary antibody (dilution 1:1000) was added for the next 1 h. During this step, the plate was covered from the light. When the secondary antibody solution was removed, the plate was washed 3 times with PBS and the wells were filled with 100 µL of PBS containing 2 µg/mL of Hoechst 33342 for the nuclei staining. The plate was visualized using a confocal laser scanning microscope BD Pathway 855 (Bioimager, Becton Dickinson, Franklin Lakes, NJ, USA) supported with AttoVision^TM^ 1.6 software.

## 4. Conclusions

UVA radiation at an applied dose of 10 J/cm^2^ downregulated the expression of p-AKT, mTOR and upregulated the expression of p53, caspase-3, caspase-9 and PARP, causing a significant reduction in the survival of human skin fibroblast and inducing apoptosis. The application of AmO at 0.1% and 0.15% concentrations to UVA-treated fibroblasts reduced the expression of apoptosis markers (p53, caspase-3, caspase-9 and PARP) and restored the expression of p-Akt and mTOR proteins. The molecular mechanism of this phenomenon is related to the stimulation of antioxidant processes through the activation of Nrf2. This suggests that *AmO* stimulated the antioxidant system in fibroblast cells and prevented the effects of UVA-induced oxidative stress. This suggests that AmO may find a wide range of pharmaceutical and cosmetological applications as a sun-protective substance.

## Figures and Tables

**Figure 1 ijms-24-10795-f001:**
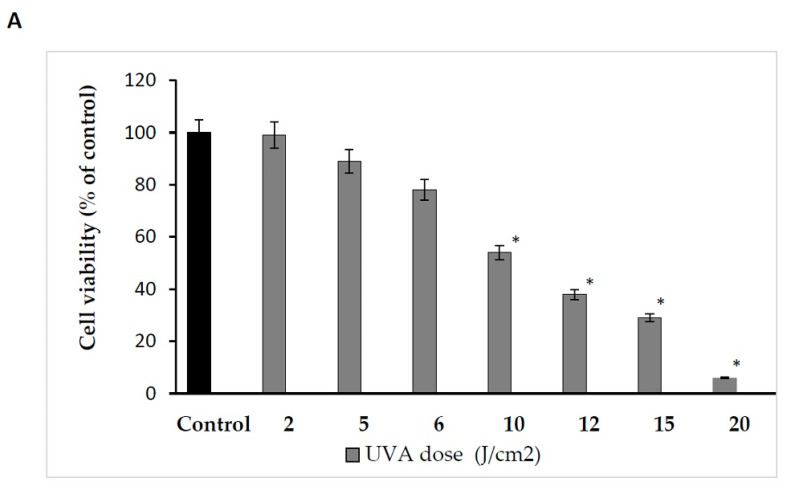
(**A**) The effects of different doses of UVA radiation on human skin fibroblasts’ cell viability. The mean ± standard error (SEM) values from the experiments performed in triplicates. * Statistically significant difference at *p* < 0.05 compared with the control. (**B**) The effects of different doses of AmO in human skin fibroblasts’ cell viability. The mean ± standard error (SEM) values from experiments performed in triplicates. * Statistically significant differences at *p* < 0.05 compared with the control.

**Figure 2 ijms-24-10795-f002:**
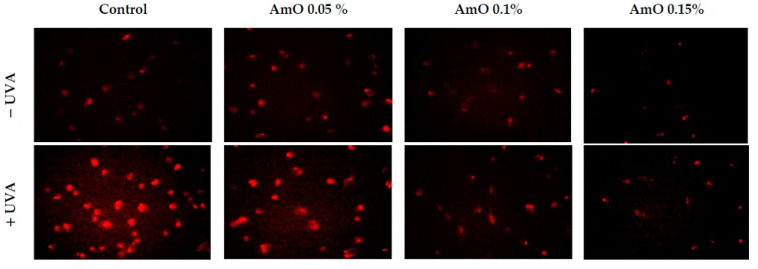
Fluorescence analysis of ROS generation in UVA irradiated fibroblasts treated with AmO at concentrations of 0.05%, 0.1% and 0.15% vs. the control. Red fluorescence intensity represents the amount of generated ROS. The images were obtained at a 20× magnification.

**Figure 3 ijms-24-10795-f003:**
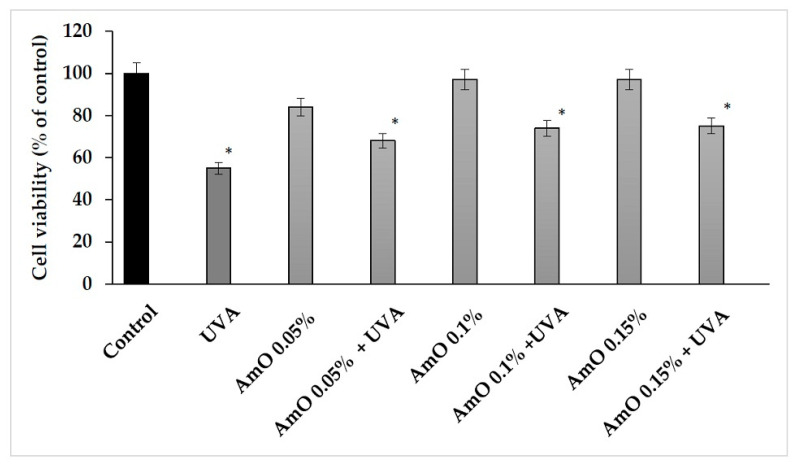
Cell viability in fibroblasts irradiated with UVA and treated with AmO at concentrations of 0.05%, 0.1% and 0.15% vs. the control. The mean ± standard error (SEM) values from the experiments performed in triplicates. * Statistically significant differences at *p* < 0.05 compared with the control.

**Figure 4 ijms-24-10795-f004:**
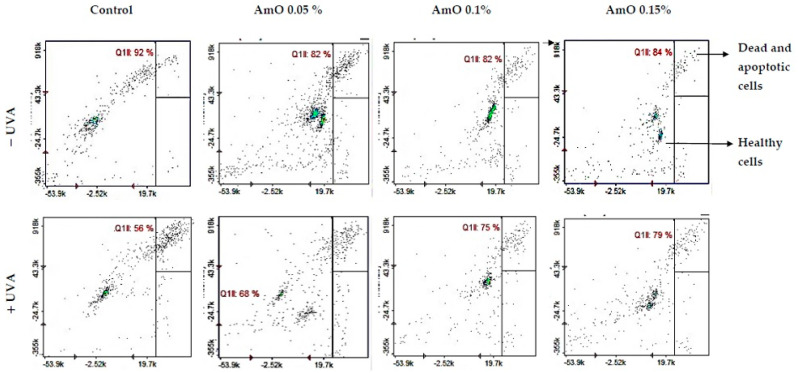
Cytometric assay of apoptosis in UVA-irradiated fibroblasts treated with AmO at concentrations of 0.05%, 0.1% and 0.15% vs. the control. The cells were stained with fluorescent dyes using Propidium Iodide and Annexin V.

**Figure 5 ijms-24-10795-f005:**
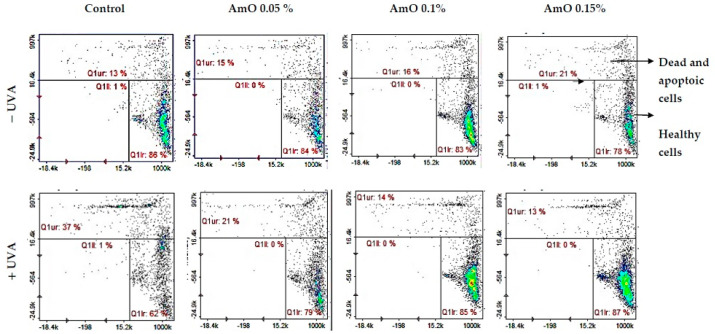
Cytometric assay of reduced thiols level in UVA-irradiated fibroblasts treated with AmO at concentrations of 0.05%, 0.1% and 0.15% vs. the control. The cells were stained with fluorescent dyes Propidium Iodide and VB-48^TM^.

**Figure 6 ijms-24-10795-f006:**
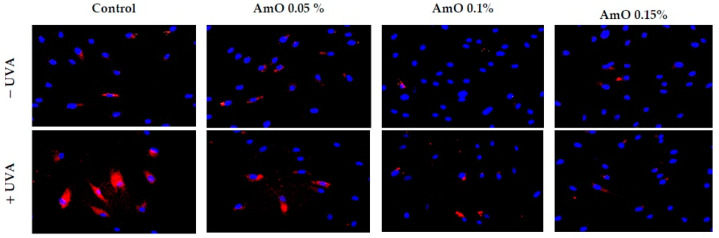
Immunofluorescence staining of p53 expression in fibroblasts irradiated by UVA in the presence of AmO at concentrations of 0.05%, 0.1% and 0.15% vs. the control. Blue staining indicates the nuclei and red staining represents p53 expression. The images were obtained at a 20× magnification.

**Figure 7 ijms-24-10795-f007:**
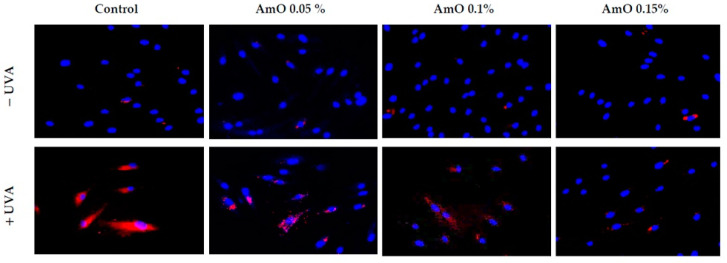
Immunofluorescence staining of caspase-3 expression in UVA-irradiated fibroblasts in the presence of AmO at concentrations of 0.05%, 0.1% and 0.15% vs. the control. Blue staining indicates the nuclei and red staining represents caspase-3 expression. The images were obtained at 20× magnification.

**Figure 8 ijms-24-10795-f008:**
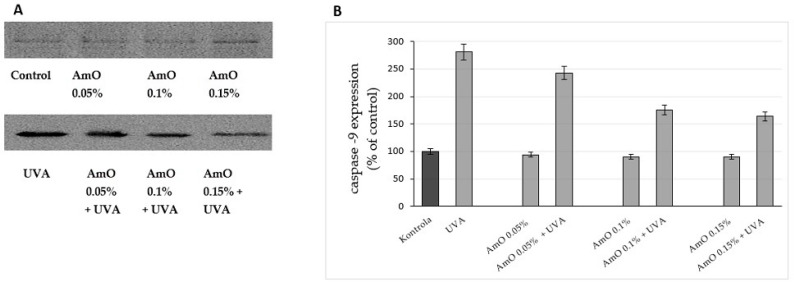
Western blot analysis (**A**) and densitometry results via ImageJ^®^ (**B**) for caspase-9 expression in UVA-irradiated fibroblasts in the presence of AmO at concentrations of 0.05%, 0.1% and 0.15% vs. the control. The mean values of 3 pooled cell homogenate extracts from 3 independent experiments are presented.

**Figure 9 ijms-24-10795-f009:**
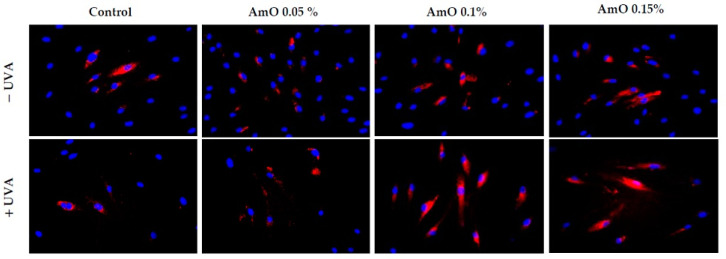
Immunofluorescence staining of p-Akt protein expression in UVA-irradiated fibroblasts in the presence of AmO at concentrations of 0.05%, 0.1% and 0.15% vs. the control. Blue staining indicates the nuclei and red staining represents p-Akt protein expression. The images were obtained at 20× magnification.

**Figure 10 ijms-24-10795-f010:**
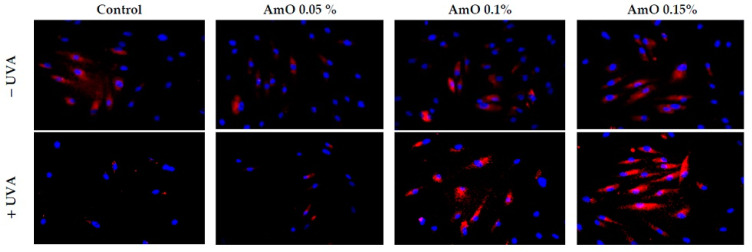
Immunofluorescence staining of mTOR protein expression in UVA-irradiated fibroblasts in the presence of AmO at concentrations of 0.05%, 0.1% and 0.15% vs. the control. Blue staining indicates the nuclei and red staining represents mTOR protein expression. The images were obtained at 20× magnification.

**Figure 11 ijms-24-10795-f011:**
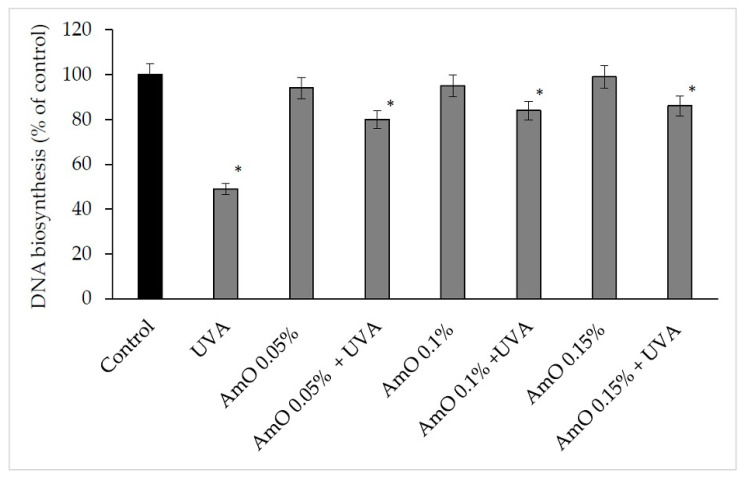
DNA biosynthesis in the UVA irradiated fibroblasts in the presence of AmO at concentrations 0.05%, 0.1% and 0.15% vs. the control. The mean ± standard error (SEM) values from the experiments performed in triplicates. * Statistically significant differences at *p* < 0.05 compared with the control.

**Figure 12 ijms-24-10795-f012:**
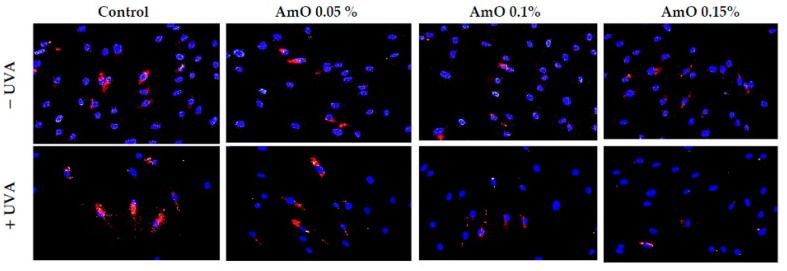
Immunofluorescence staining of PARP protein expression in UVA-irradiated fibroblasts in the presence of AmO at concentrations of 0.05%, 0.1% and 0.15% vs. the control. Blue staining indicates the nuclei and red staining represents PARP protein expression. The images were obtained at 20× magnification.

**Figure 13 ijms-24-10795-f013:**
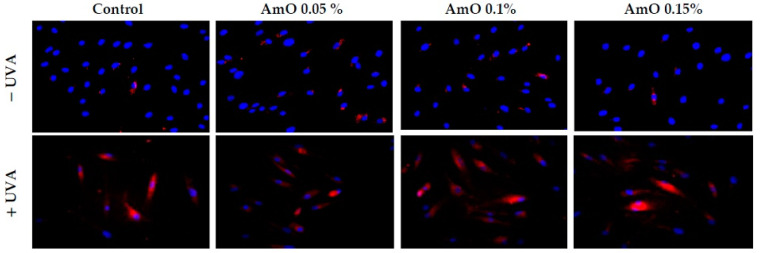
Immunofluorescence staining of Nrf2 protein expression in UVA-irradiated fibroblasts in the presence of AmO at concentrations of 0.05%, 0.1% and 0.15% vs. the control. Blue staining indicates the nuclei and red staining represents Nrf2 protein expression. The images were obtained at 20× magnification.

**Table 1 ijms-24-10795-t001:** The MTT test for cell viability in fibroblasts irradiated with UVA and treated with AmO at concentrations of 0.05%, 0.1% and 0.15% vs. the control. The mean values, standard deviation (SD), standard error (SEM) and median (Me) from the 3 experiments. * Statistically significant differences at *p* < 0.05 compared with the control. ** Statistically significant differences at *p* < 0.05 compared with the UVA group.

Non-UVA-Treated Cells								
	I	II	III	Mean Value	% Value	SD	SEM	Me
Control	0.794	0.789	0.74	0.774	100	0.029838	0.023	0.789
AmO 0.05%	0.74	0.685	0.542	0.655	84.711	0.102207	0.076	0.685
AmO 0.1%	0.78	0.76	0.709	0.749	96.856	0.036611	0.027	0.76
AmO 0.15%	0.804	0.731	0.721	0.752	97.157	0.04531	0.035	0.731
**UVA-Treated Cells**								
UVA (+)	0.449	0.413	0.425	0.429	55.426 *	0.01833	0.013	0.425
AmO 0.05%	0.564	0.496	0.51	0.523	67.614 *	0.035907	0.027	0.51
AmO 0.1%	0.461	0.636	0.625	0.574	74.160 */**	0.098015	0.075	0.625
AmO 0.15%	0.51	0.672	0.554	0.578	74.763 */**	0.083769	0.062	0.554

**Table 2 ijms-24-10795-t002:** The dpm values of radioactive [methyl-^3^H]thymidine incorporation into DNA in UVA-irradiated fibroblasts in the presence of AmO at concentrations of 0.05%, 0.1% and 0.15% vs. the control. The mean values from 3 experiments. * Statistically significant differences at *p* < 0.05 compared with the control.

Non-UVA-Treated Cells											
	I	II	III	I	II	III	I	II	III	Mean Value	% Value
Control	4411	4316	4363	3991	3556	3639	4039	4058	4049	4046.8	100
AmO 0.05%	3951	3740	3709	3513	3556	3534	3977	3895	3936	3756.7	92.85
AmO 0.1%	3808	3564	3532	3862	4265	3910	3688	4064	3710	3822.5	94.47
AmO 0.15%	4000	4278	4292	3816	3622	3719	3732	4142	4098	3966.5	98.03
**UVA-Treated Cells**											
UVA (+)	1937	1833	1944	2616	2253	2326	2371	2301	2336	2213	54.69 *
AmO 0.05%	2838	2828	2833	2838	2826	2832	2542	2525	2533	2732.7	67.54 *
AmO 0.1%	3421	3501	3461	3228	3214	3221	3622	3425	3462	3395	83.91 *
AmO 0.15%	3359	3342	3350	3950	4038	3994	3241	3175	3208	3517.4	86.93 *

## Data Availability

The data presented in this study are available in the main text of this article or on request from the corresponding author.

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
