# Peer review of "Protective Effect of Amaranthus cruentus L. Seed Oil on UVA-Radiation-Induced Apoptosis in Human Skin Fibroblasts"

_ijms, 2023, doi:10.3390/ijms241310795_

Round 1

Reviewer 1 Report

The authors have well-written review regarding the Protective effect of Amaranthus cruentus seed oil on the UVA radiation-induced apoptosis in human skin fibroblasts. The article is a comprehensive and well-designed study. I hope the article will be interesting to scientific community and adds sufficient value to the literature regarding skin damage by UVA. I may pasting here the minor comments that the authors can follow to make it easier for the readers to follow it.

  1. English language should be improved.
  2. Authors should proof read the article as there a number of typographical mistakes throughout the article. Such as in Method section (3.2.1. Cell lines and culture) line number 320 there is redundancy as “and maintained were maintained”
  3. Authors should clarify the name of the studied plant weather it is Amaranthus cruentus or  Amaranthus cruentus L.?
  4. The abstract section is a bit long, it should be concise.
  5. In introduction section, authors should elaborate more on Amaranthus cruentus regarding its taxonomy, chemical constituents, conventional uses and current research on its pharmacological potential.   
  6. In material and method section, authors should mention that seeds of which variety (wild or domestic) of Amaranthus cruentus were used for oil production.
  7. In material and method section, authors mentioned the fatty acid profile and composition of Amaranthus cruentus oil. Give its reference.  
  8. What are future prospects and the impact of current study on cosmetics and its cosmeceutical applications? 

A minor rivision of English is required  

Author Response

Answer for the Reviewer 1

The authors have well-written review regarding the Protective effect of Amaranthus cruentus seed oil on the UVA radiation-induced apoptosis in human skin fibroblasts. The article is a comprehensive and well-designed study. I hope the article will be interesting to scientific community and adds sufficient value to the literature regarding skin damage by UVA. I may pasting here the minor comments that the authors can follow to make it easier for the readers to follow it.

  1. English language should be improved.
  2. Authors should proof read the article as there a number of typographical mistakes throughout the article. Such as in Method section (3.2.1. Cell lines and culture) line number 320 there is redundancy as “and maintained were maintained”

Answer: The English language was proof read by senior scientist and typographical mistakes were corrected.

  1. Authors should clarify the name of the studied plant weather it is Amaranthus cruentus or  Amaranthus cruentus L.?

Answer: The name of the plant was corrected into Amaranthus cruentus L. in entire manuscript.

  1. The abstract section is a bit long, it should be concise.

Answer: The abstract section was shortened.

  1. In introduction section, authors should elaborate more on Amaranthus cruentus regarding its taxonomy, chemical constituents, conventional uses and current research on its pharmacological potential.

Answer: The text in lines 63-94 has been improved and supplemented with information on Amaranthus cruentus L. taxonomy, chemical constituents, conventional use and current research on its pharmacological potential.

  1. In material and method section, authors should mention that seeds of which variety (wild or domestic) of Amaranthus cruentus were used for oil production.

Answer: The sentence in line 315-317 was supplemented with data on Amaranthus cruentus origin: “It is an unrefined oil obtained via a seed cold-pressing process from locally cultivated Amaranthus cruentus species.”

  1. In material and method section, authors mentioned the fatty acid profile and composition of Amaranthus cruentus L. oil. Give its reference.

Answer: The plant material was purchased directly from its manufacturer “SzarÅ‚at M. i W. Lenkiewicz s.j. Zawady, Poland”, the domestic company. The Material Safety Data Sheet for tested oil from Amaranthus cruentus L. seed confirms its purity and composition. The document is attached to this article in supplementary material to clarify the given profile.

  1. What are future prospects and the impact of current study on cosmetics and its cosmeceutical applications?

Answer: This report provide rational for further study (animal, clinical study) on possible application of AmO in pharmacy and cosmetology as a sun protective-preparation. However, our main interest is to establish the role of proline dehydrogenase/proline oxidase (PRODH/POX), a mitochondrial enzyme, in generation of ROS in melanoma cells submitted to UVA radiation. The presented studies showing the protective effect of AmO on fibroblast damage caused by UVA radiation provide knowledge on how to protect normal (non-cancerous) tissues against the effects of this radiation. Therefore, our future plans are related to the application for a grant on experimental pharmacotherapy of melanoma.

We thank you the Reviewer for all those insightful comments.

Reviewer 2 Report

Thank you for giving me the chance to review the manuscript [Protective effect of Amaranthus cruentus seed oil on the UVA radiation-induced apoptosis in human skin fibroblasts]. It was very interesting, and with some work, the document will be a valuable contribution to the biology field. The results were very interesting, the data was abundant, and the explanations were awesome. However, the manuscript needs some overall revision in its format and sentences. Also, after revising the whole grammar and English styles, this research will benefit biology researchers.

Author Response

Answer to the Reviewer 2

Comment and Suggestions for Authors

This manuscript describes the “Amaranthus cruentus seed oil and their effects on UVA-induced apoptosis in human skin fibroblast”. The focus of this manuscript deserves attention and would provide useful information for readers. However, the manuscript slightly lacks appropriate organization, because the experiments and findings have not been explained in a right form, please consider further the following points:

General points

  1. There are many errors in the notation of Amaranthus cruentus seed oil, the main sample. Please write the scientific name in italics, and it would be better to write it in abbreviation ‘AmO’ in the text.

Answer: The name of Amaranthus cruentus L. seed oil was corrected in the entire manuscript and the abbreviation “AmO” was introduced.

  1. Introduction shows a convincing reason why the sample was selected, but this study's purpose seems insufficient. As explained in the Conclusion, it would be good to add one or two lines of research purpose, focusing on ways that can be ultimately used through research.

Answer: The comment was addressed in the “Introduction” section of revised manuscript (red labeled paragraph).

  1. The manuscript contains a large amount of information which makes the manuscript excellent. But some of the sentence is too long, which may harm the flow of the text and make it difficult to understand. Rather than putting all the information in one sentence, picking up only the key points and dividing them into several short and concise sentences would be better.

Answer: The English language and structure of sentences were improved.

  1. There are some parts where the punctuation mark or notation is incorrect. Please check the whole manuscript again.

Answer: The punctuations and notations were corrected in the entire manuscript.

  1. Positive control is very helpful in comparing the efficacy of the main sample or improving the overall completeness of the experiment. Please explain why the writers did not set up the positive control and experimented together with AmO.

Answer: We suggest that protective effect of AmO on UVA-induced damage of skin fibroblasts is a result of action of several constituents of the preparation. Therefore, we could not find appropriate positive control. However, the comment is of interest and will be addressed in our future study.

  1. The mechanism of UVA stimulating apoptosis and causing oxidative stress is also well described, but there seems to be a little lack of clear explanation of the connection between the two. Please add other factors, or references, that can weave the two mechanisms together.

Answer: The ROS dependent apoptosis is well known phenomenon [17]. Citation was added into revised manuscript.

(17)     Arfin, S.; Jha, N. K.; Jha, S. K.; Kesari, K. K.; Ruokolainen, J.; Roychoudhury, S.; Rathi, B.; Kumar, D. Oxidative Stress in Cancer Cell Metabolism. Antioxidants 2021, 10 (5), 642. https://doi.org/10.3390/antiox10050642.

  1. The style and grammar in the document should be improved. Meticulous scrutiny should be performed throughout the manuscript.

Answer: The English language was proof read by senior scientist and grammar and typographical mistakes were corrected.

Specific points

  1. Keywords: Please change ‘Amaranthus cruentus’ into italic.

Answer: correction was made.

  1. The keywords ‘pharmacy’ and ‘cosmetology’ is only mentioned in abstract and conclusion. Please refer General points No. 2 and add the contexts in Introduction.

Answer: This comment was addressed in “Introduction” section of revised manuscript (red labeled sentences at the end of introduction section).

  1. Introduction: Line 43-46 needs appropriate references.

Answer: The appropriate reference was added (line 47)

(3) Bernerd, F.; Asselineau, D. UVA Exposure of Human Skin Reconstructed in Vitro Induces Apoptosis of Dermal Fibroblasts: Subsequent Connective Tissue Repair and Implications in Photoaging. Cell Death Differ. 1998, 5 (9), 792–802. https://doi.org/10.1038/sj.cdd.4400413.

  1. Line 64-68: There is too much information, and same content is repeated. It can be shortened like: ‘Seed oil from Amaranthus cruentus L. is one of the potential sources of such a compound, as it contains abundant active substances. Among them, linoleic acid is well-known as antioxidant.’

Answer: This comment was addressed in the revised manuscript.

  1. Line 77-79, Line 91-93: Same suggestion with point 4 above.

Answer: This was corrected at points: 4-5:

Text in lines 64-95 has been changed and improved due to suggestion of the Reviewer 1: "In introduction section, authors should elaborate more on Amaranthus cruentus regarding its taxonomy, chemical constituents, conventional uses and current research on its pharmacological potential.”

  1. Line 90: ‘IC50’ into ‘IC50’.

Answer: correction was made.

  1. Line 93: ‘including UV radiation deplete’ into ‘including UV radiation, deplete’.

Answer: correction was made.

  1. Line 95: Same comment with point 1 above.

Answer: correction was made.

  1. Line 91-109: All scientific names should be changed into italic.

Answer: correction was made.

  1. In the overall manuscript, the sample names are all written in full scientific name. This seems to be the main reason for the sentences to be long. Since Line 312 has the abbreviation ‘AmO’, why don't the writer mention (AmO) after the sample in Line 66 and abbreviate all the sample names after that? AmO is also written in all Figures, but not much in the main context.

Answer: correction was made. All scientific names were correct in italics, and the abbreviation AmO is used in the text.

  1. Line 95-97: Overlapped information with introduction. No need to write down all the contents of AmO.

Answer: correction was made.

Corrected points: 4-5: Text in lines 67-95 has been changed and improved because of the suggestion of Reviewer 1: "In introduction section, authors should elaborate more on Amaranthus cruentus regarding its taxonomy, chemical constituents, conventional uses and current research on its pharmacological potential.

  1. Figure 1. (Line 111-113): Can be better if the sentence is revised. ‘The effect of UVA radiation (A) and different doses of AmO (B) in human skin fibroblasts’ cell viability.

Answer: correction was made (line 114-115)

  1. Line 113: Repeated phrase. ‘from the 3 experiments, performed in triplicates,’.

Answer: correction was made.

  1. Line 114, Line 258, Line 269: ‘* p < 0.05’ into ‘* p < 0.05’. Please check the whole format thoroughly the manuscript.

Answer: correction was made in the whole manuscript.

  1. Line 115: ‘Ko et al’ into ‘Ko et al.’. Please check the whole format thoroughly the manuscript.

Answer: correction was made.

  1. Line 115-119: It would be more persuasive to put this information in the paragraph describing Figure 1B above, and to say it as the reason for setting the concentration of the subsequent experiment.

Answer: correction was made.

Corrected: Lines 115 -199 has been put in the paragraph above Figure 1B (line 124-128) and also to clarify the results, the “cell viability” section has been divided into 2 parts including the division of Figures 1A and 1B. Lines 105-116 with Figure 1A refer to the effect of different doses of UVA radiation on fibroblast survival while lines 117-130 with Figure 1B refer to fibroblast survival after incubation with different concentrations of amaranth seed oil.

  1. Line 126: Please erase the phrase ‘oil’.

Answer: correction was made.

  1. Line 127-130, Line 300-302: If the information about sea buckthorn is for comparison between samples, please make it accurate (by the % or specific numbers), and if it's not, I'd like to know why the writer put it in.

Answer: correction was made.

  1. Line 134: All scientific names should be changed into italic.

Answer: correction was made.

  1. Line 139, Line 143, Line 149: Please check the whole format thoroughly the manuscript.

Answer: correction was made.

  1. Figure 3: In AmO 0.05%, the cell viability is lower than the other two doses. There is no UVA irradiation, and has the lowest AmO concentration, so why is the cell survival the lowest? As explained earlier in the manuscript, shouldn't the cytotoxicity appear at high concentrations and the remaining concentrations be similar?

Answer: Viability test with different concentrations of AmO has been performed several times in separate experiments and the results were similar (e.g. Figure 1B and Figure 3). Although in 0.05% AmO treated fibroblasts the cell viability was lower than at the other two doses, the differences versus control were not statistically significant.

  1. Line 153-154: Please match the format with Figure 3 by adding bracket.

Answer: correction was made.

  1. Line 171-172: The phrase ‘non-UVA-treated (UVA -) and UVA-treated (UVA +)’ is too long. The writer can mention this at the front and change all the words into only (UVA -) and (UVA +), or stick to the form of Figure 1-3 and change as (UVA -) into =control.

Answer: correction was made in the whole manuscript.

  1. Line 181, Line 197: Same with point 15.

Answer: correction was made.

  1. Line 197-203: Why is the content about UVB included? How about explaining the pathway of apoptosis and the specific factors described below one more time?

Answer: The information on UVB was intended to justify the interest in the action of UVA as an inducer of oxidative stress. Although UVB also induces apoptosis, it does not occur through ROS. The sentences on UVB were therefore to constitute arguments for the thesis that antioxidant substances can counteract the destructive effect of UVA on skin fibroblasts.

  1. Figure 8: There is no mention about experiments being performed in triplicates.

Answer: correction was made. The mean values of 3 pooled cell homogenates extracts from 3 independent experiments are presented. 

  1. Figure 8: The band is too dark, making it hard to confirm the difference in protein expression. The writer should get the density value and add appropriate graph showing the difference.

Answer: The densitometry has been performed using ImageJ tool and a graphic has been improved showing the differences (Figure 8B, page 8).

  1. 28. Line 253: seed ‘oil’.

Answer: correction was made (line 259)

  1. Figure 12: The result is magnificent, but there’s some problem finding the difference between PARP protein expressions. Please find the other way to evaluate the protein, or the writer’s immunofluorescence result so that the reader can understand the results intuitively.

Answer: The quality of the photos has been improved.

  1. 30. Line 61: In the manuscript, abbreviations must be written with all names unpacked when they are first mentioned. Write the full name of Nrf2.

Answer: In line 62 full name was written for NRF2- Nuclear factor-erythroid 2 related actor 2

  1. 3.1.1. Plant materials: It says that the sample was purchased commercially. Is there a basis to be sure if the purchased sample is really AmO? If not, the writer must add the analysis data (HPLC, Q-Tof…) that confirms the sample contains indicator components of Amaranthus cruentus seed.

Answer: The plant material was purchased directly from its producer “SzarÅ‚at M. i W. Lenkiewicz s.j. Zawady Polska” company based in Poland, we asked to provide the Material Safety Data Sheet for tested oil from Amaranthus cruentus seed so we would have no doubt as to its purity and composition. The safety data sheet is attached to this article in supplementary materials to clarify the given profile.

  1. Line 350, Line 364, Line 411: The exponent for cell number is miswritten. In addition, even if the exponent is accurately marked, the number of cells used in the experiment seems to be too small. What are the criteria for calculating the number of cells?

Answer: The cells were counted manually using hemocytometer. The cell number was corrected into 3 × 105 cells/well of 6 well plate. It was initial number of cells that were plated. During growth, the number of cells increased about 5 times, close to confluency at about 1.5 × 106 cells/well. Such number of cells was used in the experiments.

  1. Line 383: Huge blank placing in the middle of the sentence.

Answer: correction was made.

Thank you for giving the chance to review the manuscript [Protective effect of Amaranthus cruentus seed oil on the UVA radiation-induced apoptosis in human skin fibroblasts]. It was very interesting, and with some work, the document will be a valuable contribution to the biology field. The results were very interesting, data was abundant, and explanations were awesome. However, the manuscript needs some overall revise on its format and sentences. Also, after revising the whole grammars and English styles, this research will be a good asset to biology researchers.

Round 2

Reviewer 2 Report

I think a lot of things have been improved. Thank you for your efforts.